# Biased β-Agonists Favoring Gs over β-Arrestin for Individualized Treatment of Obstructive Lung Disease

**DOI:** 10.3390/jpm12030331

**Published:** 2022-02-22

**Authors:** Alina Tokmakova, Donghwa Kim, William A. Goddard, Stephen B. Liggett

**Affiliations:** 1Program in Biophysics, University of California, San Francisco, CA 94102, USA; alina.arzamasskaia@ucsf.edu; 2Department of Medicine, Morsani College of Medicine, University of South Florida, Tampa, FL 33612, USA; donghwa@usf.edu; 3Center for Personized Medicine and Genomics, Morsani College of Medicine, University of South Florida, Tampa, FL 33612, USA; 4Materials and Process Simulation Center, California Institute of Technology, Pasadena, CA 91125, USA; wag@caltech.edu; 5Department of Molecular Pharmacology and Physiology and Department of Medical Engineering, Morsani College of Medicine, University of South Florida, Tampa, FL 33612, USA

**Keywords:** β_2_-adrenergic receptor, tachyphylaxis, G protein, desensitization, airway smooth muscle, G-protein-coupled receptor kinases

## Abstract

Signals from G-protein-coupled receptors (GPCRs) are the most frequently targeted pathways of currently prescribed therapeutics. Rather than being a simple switch, it is now evident that a given receptor can directly initiate multiple signals, and biasing to achieve signal selectivity based on agonist structure is possible. Biased agonists could direct therapeutically favorable pathways while avoiding counterproductive or adverse reaction pathways. For obstructive lung diseases, β_2_-adrenergic receptor (β_2_AR) agonists act at these receptors on airway smooth muscle (ASM) cells to open the airways by relaxing ASM, improving airflow and morbidity. However, these receptors signal to the G protein Gs (increasing cAMP and promoting relaxation), but also to β-arrestin (promoting desensitization and a loss of effectiveness). Indeed, β-agonist use is associated with adverse events in asthma pathogenesis and clinical outcomes which are related to desensitization. β-agonists favoring Gs coupling over β-arrestin binding would provide a means of tailoring bronchodilator therapy. In this review, we show how combinatorial methods with a 40 million compound agnostic library led to a new class of biased β-agonists that do not desensitize, providing an opportunity to personalize therapy in patients who experience poor efficacy or adverse effects from traditional balanced agonists.

## 1. Introduction

G-protein-coupled receptor (GPCR) proteins represent the largest superfamily of proteins in the body, each with seven transmembrane helices spanning the lipid bilayer. GPCRs are expressed on every cell type, and respond to autocrine, paracrine and endocrine hormones, neurotransmitters, metabolites, chemokines, and many other molecules. Approximately 50% of prescribed drugs target GPCR signaling networks, acting as agonists or antagonists. For the non-visual GPCRs, the amino-terminus is extracellular and the carboxy-terminus is intracellular. The transmembrane domains (TMDs) are connected by three extracellular loops (ECL1-3) and three intracellular loops (ICL1-3). For Class A GPCRs, agonist binding occurs within a pocket formed by the upper portions (extracellular ends) of the TMDs and the ECLs. GPCRs transduce signaling by coupling to heterotrimeric G proteins, consisting of α, β, and γ subunits. In the traditional model of signaling, agonist activation results in the binding of the receptor to the α subunit of the G protein, with the subsequent dissociation of the heterotrimer into the α and βγ subunits. The α-subunit then activates (or inactivates) an effector enzyme, such as the activation of adenylyl cyclase, which occurs with receptors that couple to Gαs. Activated adenylyl cyclase converts adenosine triphosphate (ATP) to cyclic adenosine monophosphate (cAMP, termed the “second messenger”). cAMP activates protein kinase A, which phosphorylates numerous proteins resulting in the biochemical or physiological response of the cell. The dissociated βγ subunits may also serve to evoke signaling. The termination of the cycle occurs by the reassociation of the heterotrimer; the cycle repeats as long as the agonist remains present. Most GPCRs display a rapid attenuation of signaling with continuous agonist activation, termed desensitization, which is thought to be a mechanism of fine-tuning the response in cells that are receiving hundreds of receptor signals [1]. Desensitization may limit the therapeutic effectiveness of administered GPCR agonists, which is clinically also referred to as tachyphylaxis or agonist subsensitivity. The most rapid form of agonist-promoted desensitization is mediated by the phosphorylation of intracellular Ser or Thr by G-protein-coupled receptor kinases (GRKs). The GRK-phosphorylated receptor is a substrate for the binding of arrestins, which interdict between receptor and G protein, causing a partial loss of signaling. As will be further discussed in the context of biased agonism, arrestins also promote additional regulatory events during more prolonged agonist activation, and they evoke de novo signals that are independent of the G protein.

## 2. Biased Agonist Signaling at GPCRs

For decades, GPCRs were thought to activate a single pathway (such as cAMP production) in a simple “on–off” manner. However, by 1992 we had shown that a single receptor could couple to two different G proteins [2], leading to multifunctional signaling. In the case of the α_2A_-adrenergic receptor, the structural determinants of Gs and Gi coupling were mapped to the third ICL of the receptor [3]. Subsequently, we showed that α_2_AR agonists of different structures could bias the receptor to preferentially activate the Gs or the Gi pathway [4]. We illustrated this action as shown in Figure 1A, where agonist A was balanced, with the agonist-bound receptor adopting a conformation denoted AR* that coupled to both G proteins, while agonist B promoted the adoption of conformation BR^∆^, which preferentially activated one G protein over the other. Although not termed “biased agonism” at that time (1994), to our knowledge this was the first description of the biased phenomenon with GPCRs.

Biased agonism has also been termed pathway selectivity and functional selectivity. For the purposes of this review, a biased agonist is defined as an agonist that preferentially activates one pathway over another. A balanced (unbiased) agonist activates both pathways with similar potencies and efficacies. The definition can be extended to include multiple pathways, where the biased agonist differentially activates, or does not activate, one or more pathway subsets of the full repertoire. As such, the earlier depiction of pathway selectivity (Figure 1A) can be expanded as shown in Figure 1B,C. In this case, in the non-agonist bound state the GPCR “oscillates” through many active conformations (Figure 1B) that evoke signals (albeit weakly in the non-agonist bound state); an agonist can stabilize a select few, or one, pathway with greater efficacy or potency than other pathways. This would not necessarily exclude some degree of signaling by the less efficient pathways by an apparently biased agonist, but the signal would be minimal compared to the forwardly biased pathway (see Figure 1C and legend). The testing of potentially biased agonists requires parallel experiments with a balanced agonist, since the various assays utilized for measuring the signals usually exhibit some degree of “system bias”, necessitating a comparison with a known unbiased agonist. System bias can be a particularly vexing problem when the assay for one pathway is not particularly robust or sensitive, or has a low signal-to-noise ratio, compared to the assay used for the second pathway. This subset of system bias we term “assay bias”, and when present, it can give the impression of true ligand biasing because of the differences in pathway detection assays. Under these circumstances, at most doses, the one pathway appears not to be activated, while the other does appear to be activated, suggesting bias. Another type of bias, which we term “intrinsic bias”, exists when the efficiency of activation by one pathway is significantly different from the other pathway. In this case, a balanced agonist may not be possible since the two pathways differ intrinsically as to their efficiency of activation. Experiments where there is assay bias or intrinsic bias may be even more difficult to interpret when the compound under investigation is a partial agonist (i.e., has low intrinsic activity for the primary pathway) [5]. Note that true agonist biasing can occur due to the relative loss of signaling to one pathway compared to another, or a relative gain of signaling by one pathway while the other signaling remains the same. As of this writing, a loss of function of one pathway has been typically reported as the basis of biased agonist activity at GPCRs. Defining a therapeutically useful biased agonist is dependent on identifying the pathway(s) that is(are) desired, or those that are considered deleterious, for a given clinical scenario. As such, it is important to ascertain the biochemical or physiological outcome of a potentially biased agonist in the cell type of interest with the endogenous expression of receptors and associated proteins, such as the GRKs and β-arrestins.

Several mechanisms have been identified as the basis of biasing in multifunctional GPCR signaling. Some receptors couple to two G proteins, and such coupling can be selectively activated by structurally distinct agonists. As introduced earlier, many GPCRs also adopt agonist-stabilized conformations that promote the phosphorylation of the ICL3 or the C-terminus (CT) by GRKs. This leads to the binding of β-arrestin1 or β-arrestin2 to these regions, followed by a second set of interactions within the core of the receptor, which compete for G protein binding [6,7]. This effect, sometimes called “uncoupling”, is the major mechanism of agonist-promoted homologous desensitization. The initial GRK phosphorylation requires Ser or Thr as phosphoacceptors and may depend on the extent of the expression of a given GRK isoform [8]. However, the conformation of the TMDs as stabilized by agonist, and then transmitted to these intracellular regions, is a critical determinant of which residues are phosphorylated [9]. Collectively, these events represent a phospho-barcode [10]. Thus, based on the conformation of the ICL3 or CT, and the localization of the phosphoacceptor sites, GRK phosphorylation may differ based on the structure of the agonist, leading to a “textured” β-arrestin, which can be manipulated by agonist structure. Two obvious consequences of this variable β-arrestin action are differential desensitization and differential signaling by β-arrestin itself. This latter property is due to the chaperone and scaffold capacity of β-arrestin that brings together multiple proteins into the complex. A classic example is the receptor-mediated activation of mitogen activated protein kinase (MAPK, ERK1/2), which is dependent on the agonist-mediated binding of β-arrestin to the receptor. β-arrestin signaling can be highly cell-type dependent, based on the expression of the different β-arrestins or GRKs and the stoichiometry between receptors and these proteins.

In this review, we concentrate on the recent progress [11] in the discovery and characterization of biased agonists acting at the β_2_-adrenergic receptor (β_2_AR), a Gs-coupled receptor that is expressed on many cell types and is the target for therapeutically administered agonists for treating obstructive lung disease.

## 3. The Clinical Need for a Gs-Biased β_2_AR Agonist for Treating Obstructive Lung Disease

Biased agonists acting at many other GPCRs have been found, and some have undergone clinical trials [12]. For example, the μ-opioid receptor agonist oliceridine does not engage β-arrestin, but it maintains G protein coupling [13]. The result is potent analgesia, and, somewhat unexpectedly, decreased opiate-related adverse effects [14], although the human trials were less impressive than the rodent-based studies. An agonist acting at the angiotensin II type 1 receptor has been found, which fails to promote G protein (Gq) coupling, but it is biased towards β-arrestin recruitment [15]. Events from this latter signal normalized blood pressure and preserved cardiac output in animal models, unlike the balanced agonist acting at this receptor. For the 5-HT_1A_ receptor, agonists have been found that display strong biasing to one pathway, while other agonists show biasing towards another pathway [16]. This suggests that this receptor might be a target for several distinctly different central nervous system diseases, which can be directed by agonist structures.

In obstructive lung diseases, such as asthma, airflow through the conducting airways is limited due to active constriction of airway smooth muscle (ASM) cells caused by local actions of bronchoconstrictive substances. This constriction is a major component of the morbidity and mortality in asthma. This process can be reversed by the action of β_2_AR agonists (termed β-agonists), which activate β_2_AR on ASM, increasing intracellular cAMP via Gs coupling, and relaxing the cells, which opens the airways. In chronic obstructive pulmonary disease (COPD), which has variable components of emphysema and chronic bronchitis, there can also be a degree of reversable bronchoconstriction. The only direct bronchodilators used to treat obstructive lung diseases are β-agonists. While generally effective, their use is characterized by a high degree of inter-individual variability in important clinical outcomes, including a number of adverse effects, presenting an opportunity for β-agonists that are therapeutically more targeted compared to those currently available. Of particular concern is the decrease in effectiveness of these drugs (tachyphylaxis) and the increased sensitivity to brochoconstrictive agents, when β-agonists are administered on a chronic basis [17,18,19,20,21] In addition, poor responsiveness in moderate-to-severe asthma is not uncommon, and an increase in exacerbations, hospitalizations, and deaths have been reported with chronic use [22,23,24,25]. The variability in these responses is thought to be dependent on severity, asthma phenotype, environmental factors, and genetic and epigenetic mechanisms [26,27]. The process of GRK phosphorylation followed by β-arrestin binding begins within seconds of agonist activation and is evident after several minutes of agonist exposure to cells or tissues. Thus, β-arrestin mediated desensitization is an integral component of the response to the first dose of a β-agonist. Indeed, we have shown, in targeted transgenic mice expressing wild-type and GRK/β-arrestin null β_2_AR on ASM, a significant increase in airway relaxation in mice expressing the latter receptors during a 30 min exposure to β-agonist [28].

In addition to this acute effect leading to uncoupling, β-arrestin also mediates β_2_AR internalization, where receptors undergo endocytosis to the cell interior, effectively reducing cell surface expression [1]. These internalized receptors are cycled to degradation pathways with continued exposure to agonist on the order of hours. This process (termed downregulation) causes an overall net decrease in cellular receptors that requires ~12 h to regenerate under conditions where the agonist is no longer present. Thus, both acute and chronic efficacy of β-agonists is influenced by the actions of β-arrestin binding to the receptor. A β-agonist substantially biased away from β-arrestin, while maintaining a sufficient degree of Gs coupling, would provide for a personalized treatment approach for patients who are poorly responsive, or experience tachyphylaxis, to balanced agonists. Interestingly, a Gs-biased β_2_AR agonist had not been found through traditional medicinal chemistry prior to our studies, so we utilized a different approach to identify compounds with this favorable biasing.

## 4. Screening Millions of Compounds for Novel β-Agonists with Combinatorial Libraries

A scaffold ranking (SR) library is a mixture-based combinatorial library containing thousands of compounds in each sample, arranged by scaffold type [29,30,31]. In our attempt to find novel β-agonists [11], we used a 40 million compound library, and the initial screen consisted of 87 sample wells, which represented a large chemical space that was agnostic to known β-agonist structures, with each well having >50,000 compounds. Any well that resulted in a positive signal (see below) was further investigated with samples having exponentially fewer compounds. For the initial screen of the SR library, Chinese hamster fibroblasts (CHW-1102 cells) stably transfected to express the human β_2_AR (CHW-β_2_) and non-transfected CHW-1102 cells were employed. Non-transfected cells express no detectable β_2_AR by radioligand binding or mRNA content and show no increase in cAMP from the full, balanced, agonist isoproterenol (ISO). A positive screen from a SR sample well was defined when cAMP was stimulated over baseline in CHW-β_2_ cells but not in the non-transfected cells [11]. The results from this screen (Figure 2A) showed several positive mixture samples, particularly well 1319, which increased cAMP by ~six-fold over baseline in the transfected cells with no stimulation in the non-transfected cells. The scaffold for well 1319 is a dihydroimidazolyl-butyl-cyclic urea with three positions used for R group substitutions (Figure 2B). Subsequently, we synthesized a custom positional scanning (PS) library with all possible combinations of the three R groups, representing ~56,000 compounds segregated into 116 sample wells. An example of results from the cAMP screening of compounds with variable R_1_ groups is shown in Figure 2C. Because of the ordered nature of the R-group substitutions, structure–activity relationships could be ascertained computationally [11,32]. R_1_ substitutions as the *S-*stereoisomer were significantly more active than the analogous *R*-stereoisomer. From this deconvolution, we synthesized twenty-four individual compounds, twelve *S-* and twelve analogous *R*-stereoisomers (Figure 3A). cAMP studies were again performed with these compounds, including multiple doses [11] to establish the potency and efficacy (Figure 3B). The criteria for further characterization of these compounds were: (1) statistically significant stimulation of cAMP in CHW-β2 cells over baseline; (2) no stimulation of cAMP from non-transfected CHW cells; (3) a cAMP response from a compound that was blocked by the βAR antagonist propranolol; (4) a stereoisomer of a single compound, which showed no cAMP response compared with the parent stereoisomer; (5) concentration-response data for cAMP that could be fit to a four-parameter least-squares regression equation (sigmoid curve with R^2^ > 0.9); and (6) an average Hill coefficient of the fitted dose–response curve of 0.8–1.3. As illustrated in Figure 3B,C, some compounds had atypical dose–response curves, the R-isomer was active, or their activation of β_2_AR was not fully blocked by propranolol. Four compounds met the criteria for further studies (C1-S, C3-S, C5-S, and C6-S).

## 5. Ascertainment of Gs-Biasing with Selected Compounds

We sought agonists that displayed little or no agonist-promoted β-arrestin binding to the β_2_AR, yet still coupled to Gs, which would be a favorable profile for a non-desensitizing β-agonist for treating asthma. We performed four different experimental assays [11] to determine the extent of β-arrestin binding. A proximity ligation assay [11,33,34] in transfected HEK-293 cells (Figure 4) showed that C5-S and C6-S evoked β-arrestin binding to β_2_AR to approximately the same extent as the partial, balanced, β-agonist albuterol (ALB). In contrast, C-1S exhibited no β_2_AR: β-arrestin binding. These results were further confirmed using an enzyme complementation assay [11,35] (ECA, PathHunter, DiscoverX, San Francisco, California) and live cell imaging of β-arrestin-GFP redistribution [11,28,36] in transfected cells. These studies revealed modest β-arrestin interaction with C5-S and ALB, but no detectable agonist-promoted interaction with C1-S, even at concentrations of 100–300 μM (~100-fold greater than the EC_50_ for cAMP stimulation). We also measured Gs dissociation (representing receptor activation and Gα coupling) using bioluminescence resonant energy transfer (BRET2) in transfected cells with tagged α and γ subunits [11,37]. These confirmed the cAMP screening and dose–response studies. Finally, we assessed the agonist activation of ERK1/2 in the presence of a protein kinase A inhibitor, which has been previously shown to be mediated by β-arrestin [38]. These experiments were conducted with human ASM cells, thus representing conditions of endogenous expression of receptor, GRKs, and β-arrestins with stoichiometery present in the cell type of interest. As indicated (Figure 4B), C1-S failed to promote ERK1/2 phosphorylation, which was consistent with the three other assays. The bias factor β [35,39] was calculated from the ECA results (the β-arrestin component) and the BRET2 results (the Gs-coupling component). In this case, β < 0 indicates β-arrestin bias, whereas β > 0 indicates Gs bias. Values of 0 (or if the 95% confidence intervals cross 0) indicate a balanced (unbiased) agonist. ISO, ALB, and C5-S were found to be balanced. Since β-arrestin binding by C1-S was not detected, β could not be calculated, but is much greater than 0, indicative of Gs biasing (see Figure 4C and legend).

## 6. The Lack of C1-S Promoted β-Arrestin Interaction Correlates with Absent Physiologic Desensitization of the Human ASM Relaxation Response

We utilized magnetic twisting cytometry (MTC) to measure human ASM relaxation in response to various β-agonists. As described elsewhere [11,40,41], the technique involves tagging the cells with ferrimagnetic microbeads coated with Arg–Gly–Asp peptides ligated to integrin receptors, which when placed in a magnetic field provide a method to measure cytoskeletal stiffness. A decrease in stiffness equates to ASM relaxation. C1-S was first tested to confirm that it indeed relaxed ASM as would be expected for a β-agonist. Having confirmed this therapeutic response (~38% relaxation, Figure 5A), we then examined the potential for β_2_AR desensitization by C1-S and ALB (the positive control). Cells were incubated with vehicle, C1-S, or ALB for 30 min and for 4 h, washed, and then challenged with the full agonist ISO and the relaxation response determined in real time over the subsequent 180 s. ALB pretreatment evoked desensitization at both time points (~35% and 70%, respectively; Figure 5B,C). In contrast, C1-S pretreatment at a saturating concentration evoked no desensitization with the 30 min or the 4 h pretreatments. Additional studies measuring cAMP also revealed no desensitization [11]. These results were consistent with the biochemical studies using the PLA, ECA, GFP-β-arrestin redistribution assay, ERK1/2 activation, and cAMP studies, which all indicated the lack of C1-S-promoted β-arrestin binding to the receptor. Thus, the apparent biasing of C1-S that favors Gs-coupling over β-arrestin binding results in relaxation without desensitization, which is the favorable therapeutic response that we sought. In contrast, and consistent with clinical observations, ALB (the most commonly prescribed β-agonist for treating asthma) promoted β-arrestin binding and displayed physiologic desensitization to both short- and long-term exposures of ASM cells to this agonist.

## 7. Modeling of the Interactions of C1-S, Epinephrine, and C5-S Identifies a Potential Structural Basis of C1-S Biasing

The lack of similarity between the structures of C1-S and ISO, ALB, epinephrine, or any other known β-agonist, and the Gs biasing of C1-S suggested that C1-S would have either gains or losses of binding sites compared to balanced β-agonists that might account for its biased phenotype. To address this, molecular dynamics (MD) studies were performed using Jaguar, DarwinDock, GROMACS, and associated programs as described [11,42,43,44,45] to generate 3D structures, time-dependent interaction energies, and pharmacophores for the interactions of C1-S and C5-S with the β_2_AR complexed with Gs (representing the activated state) and with the noncomplexed receptor. These data were compared to that from the full, balanced, agonist epinephrine. For the sake of brevity, selected data are shown in this review for the agonist-activated state, and the reader is referred elsewhere for more extensive results [11]. This approach has shown a good correlation between the simulations and experimentally derived data [46,47]. The 3D structure of C1-S binding to the β_2_AR in complex with Gs is shown in Figure 6, while the pharmacophores showing comparisons of the predicted binding sites for C1-S, epinephrine, and C5-S are shown in Figure 7. Here, we use the Ballesteros and Weinstein nomenclature [48], where, after the amino acid three-letter abbreviation, the residue number is listed, followed by a superscript that indicates the TMD number and the position within the helix. C1-S binds to Asp113^3.32^ via a strong salt bridge, while hydrogen bonds formed with Ser203^5.42^ and Asn312^7.39^. An intrareceptor hydrogen bond was evident between Asp113^3.32^ and Tyr316^7.43^. A π–π stacking interaction was found between Phe193^ECL2^ and the two aromatic groups of C1-S. Comparing the interactions between the activated β_2_AR and the unbiased agonist epinephrine, we note several important differences that might represent the basis of the Gs biasing of C1-S. At Ser203^5.42^, epinephrine acts as a proton *donor*, while C1-S is a proton *acceptor*. Furthermore, besides the salt bridge with Asp113^3.32^, epinephrine also forms a hydrogen bond with Asp113^3.32^, arising from the β-carbon hydroxyl, whereas there is no analogous residue in C1-S. In common with epinephrine, C1-S interacts with Asn312^7.39^ and Phe193^ECL2^ via hydrogen bonding. However, the other interaction at Phe193^ECL2^ with C1-S (π–π stacking) is replaced by a cation-pi interaction with epinephrine arising from the epinephrine terminal amine group. Epinephrine also binds to Ser207^5.46^, Asn293^6.55^, and Phe290^6.52^, which was not found with C1-S.

The contrasting binding sites between C1-S and epinephrine are somewhat difficult to interpret given the much larger size of C1-S. However, C5-S is structurally very similar to C1-S (Figure 3A) and the experimental results indicate that C5-S is unbiased. At R_2_, C1-S has a cyclohexane while C5-S has a benzene; otherwise, the two compounds are identical. We thus modeled C5-S and compared those results with C1-S and epinephrine. The Asp113^3.32^ and Phe193^ECL2^ interactions were similar between the two. However, C5-S interacts with Ser207^5.46^ (as epinephrine), while C1-S does not. In addition, C5-S binds to Asn293^6.55^ in TMD6, while C1-S binds to Asn312^7.39^ in TMD7; epinephrine binds to both residues. Indeed, the TMD6 interaction energies are greater for C5-S than C1-S and the TMD7 interaction energies with C1-S exceed those of C5-S [11]. We expressed mutated β_2_AR, where Asn293^6.55^ was substituted with Ala, and in a different receptor, Asn312^7.39^ was substituted with Ala. The Ala293 receptor revealed markedly impaired signaling by C5-S compared to wild-type β_2_AR. In contrast, C1-S signaling was unaffected by the mutation [11]. These experimental results confirmed the aforementioned differences in the binding between C1-S and C5-S as determined by molecular modeling. The other mutation at Asn312^7.39^ resulted in severe distortion of the binding pocket, as indicated by a loss of antagonist radioligand binding and a substantial impairment of signaling by the unbiased agonist ISO. Thus, this mutated receptor was not further studied.

Several observations from these modeling studies indicate a potential mechanism for the biasing of C1-S. We note that the absence of the Ser207^5.46^ hydrogen bond with C1-S may alter the inward bulge of TMD5 that occurs with epinephrine binding, which is associated with the outward movement of TMD6. Furthermore, epinephrine also binds to Phe290^6.52^, Asn293^6.55^, and Ser207^5.46^, and the epinephrine-activated receptor has an intrareceptor bond between Ser204^5.43^ and the Asn293^6.55^. These hydrogen bonds have been proposed to be part of a polar network for this balanced agonist [49]. In contrast, in this region the C1-S–bound receptor exhibits only the Ser203^5.42^ bond, the intrareceptor bond, and a modified interaction with Phe193^ECL2^. The mutation of Ser204^5.43^ to Thr or Ala, which disrupts this network, decreases the ISO-promoted β-arrestin binding to β_2_AR, with a smaller effect on Gs coupling [49], which points to one mechanism by which C1-S is biased away from β-arrestin. Interestingly, the Phe193^ECL2^ interaction has been implicated as one important component of β-arrestin recruitment by ISO [50]. This study indicated that the “vestibule” formed over the orthosteric binding pocket by ECL2 plays a role in how agonist stabilizes receptor conformation to accommodate β-arrestin. Epinephrine and ISO bind to Phe193^ECL2^ by means of a cation-π interaction, while C1-S binding is via a π–π interaction. This π–π interaction is weaker, being described by Van der Waal’s forces, compared to the Coulombic cation-π interaction. This weaker interaction between C1-S and Phe193^ECL2^ may also contribute to the biased signaling of C1-S compared to balanced agonist, such as ISO and epinephrine. In our studies with C5-S, which is structurally similar to C1-S (Figure 3B), we noted that C5-S did not display bias. We found a reorientation of the C5-S agonist due to benzene being more exposed to solvent compared to cyclohexane, resulting in binding to Asn293^6.55^ in TMD6, pulling TMD3 and TMD6 closer in the upper part of the receptor. C1-S binds to Asn312^7.39^ in TMD7, linking TMD3 with TMD7 instead of TMD6. The net result is a shift of the C1-S position away from the TMD3-5-6 pocket that forms with C5-S, which may also contribute to the differences in the β-arrestin activation promoted by the two compounds.

## 8. Conclusions

A component of tailoring GPCR-based therapy for individual patients may involve the administration of biased agonists or antagonists that can achieve selected biological functions, while avoiding potential negative actions, as might be encountered with balanced ligands. We show in this paper how a biased agonist acting at the β_2_AR was discovered and characterized, using an agnostic approach by screening a large chemical space combinatorial library. While both β-arrestin biased and G protein biased agonists have been found for a number of other GPCRs [12,51], such biased ligands for the β_2_AR had not been identified. The Gs-biased β-agonist we discovered, which evoked no detectable β-arrestin binding, represents a prototype that could lead to an inhaled preparation for treating obstructive lung diseases, particularly asthma. Patients with poor responsiveness to balanced β-agonists, due to genetic, environmental or epigenetic mechanisms, or disease severity, would be predicted to have a more favorable response to agonists, such as C1-S, since the acute attenuation of the bronchodilating effect would not be present. Likewise, patients who display clinically relevant tachyphylaxis to chronic therapy with a balanced β-agonist would also be predicted to have an improved outcome with a biased agonist that favors Gs-coupling over β-arrestin-mediated desensitization. The potential to personalize therapy with ligands biased in specific ways, to account for certain disease phenotypes and avoiding adverse effects, represents an exciting new aspect for drug development targeting GPCRs.

## Figures and Tables

**Figure 1 jpm-12-00331-f001:**
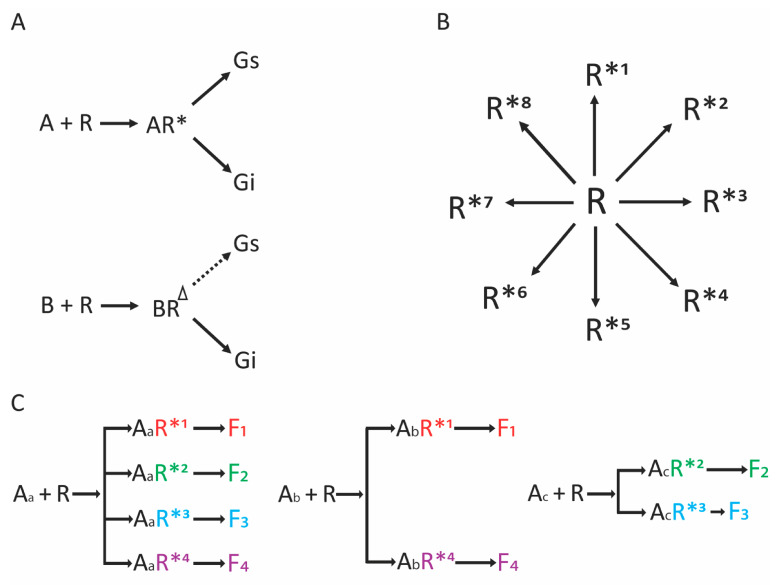
Theoretical considerations for the biasing of agonists acting at GPCRs. (**A**) An early conceptual depiction of biasing. In this case, the agonist denoted “A” interacts with the receptor (R), which results in the R* conformation yielding active coupling to two G proteins. The structurally distinct agonist “B” holds the receptor in conformation R^∆^, resulting in less coupling to Gs while maintaining Gi coupling. Modified from reference [4]. (**B**) The receptor is shown in the agonist-unoccupied form oscillating to multiple potential active conformations, denoted R*^1^–R*^8^. Generally, the signals from these conformations are minimal, since the equilibrium in the absence of agonist favors the inactive state. (**C**) Structurally diverse agonists (A_a_, A_b_, and A_c_) can stabilize only certain active conformations (R*^1^–R*^4^), leading to selective activation of a subset of potential functional signaling outcomes (F_1_–F_4_). In this example, agonist A_a_ is unbiased (also termed “balanced”), while A_b_ stabilizes two of the four conformations. This agonist is biased in favor of evoking signals F_1_ and F_4_ over F_2_ and F_3_. For agonist A_c_, the bias favors F_2_ and F_3_. However, note that the extent of stabilizing the R*^3^ conformation is diminished compared to that observed with A_a_. Thus, agonist A_c_ is biased in favor of F_2_ and F_3_, but the marginal F_3_ signal may not be physiologically evident, so from a pharmacological standpoint, A_c_ could be considered to be effectively biased only towards F_2_.

**Figure 2 jpm-12-00331-f002:**
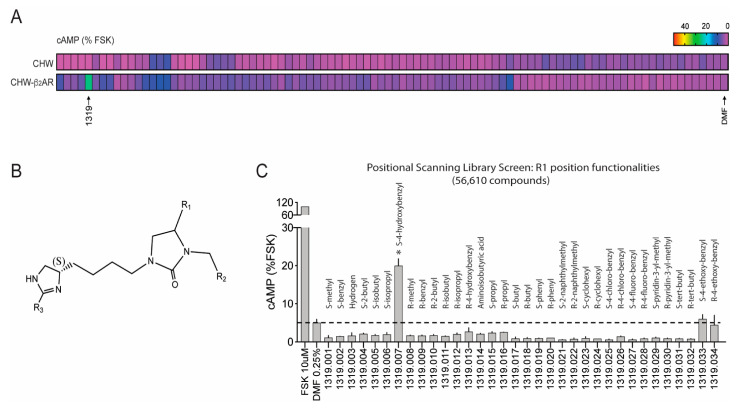
Discovery of novel β_2_AR agonists using scaffold ranking (SR) and positional scanning (PS) libraries. (**A**) The SR library consisting of samples of mixtures of chemicals was used to detect potential agonist acting at the β_2_AR. CHW cells stably transfected to express the human β_2_AR or non-transfected null cells (see text) were treated with an aliquot from each sample well and cAMP measured. cAMP values were normalized to the cAMP stimulated by the direct adenylyl cyclase activator forskolin. Results are shown as a heat map with the scale as indicated. PS library well #1319 stimulated cAMP in the β_2_AR-expressing cells to an extent greater than any other sample well. (**B**) The scaffold of well #1319 showing the positions of the three R-groups where substitutions were made. (**C**) Results from a portion of the PS library screen, where R_1_ was changed as indicated, with all other possibilities for R_2_ and R_3_. Each sample well contained 1,665 compounds. The *S*-4-hydroxybenzyl substitution at position R_1_ resulted in activity of this mixture. *, *p* < 0.01 vs vehicle (0.25% dimethylformamide [DMF]).

**Figure 3 jpm-12-00331-f003:**
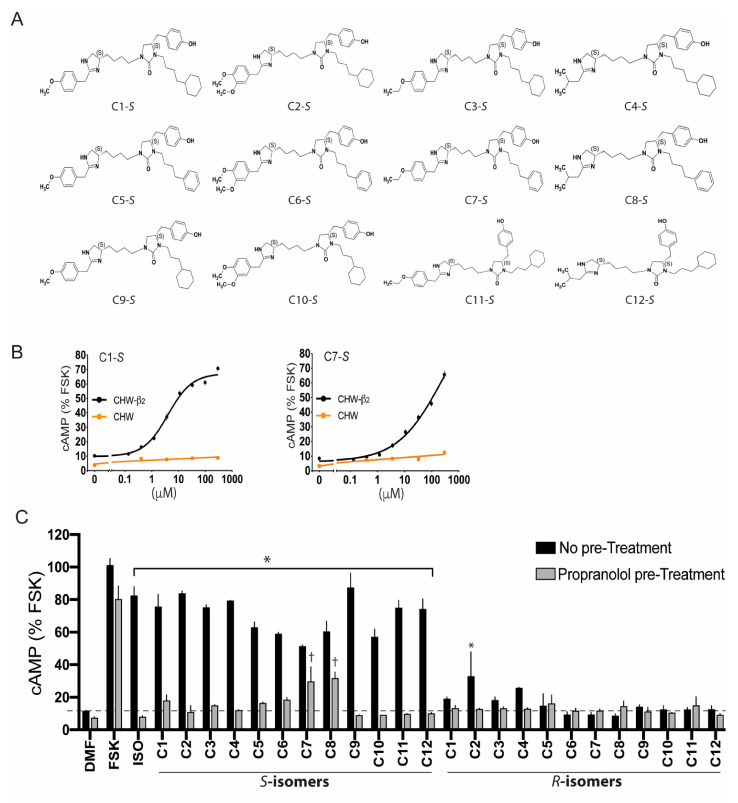
Evaluation of individual compounds derived from the discovery scaffold. (**A**) Computational methods indicated 12 specific combinations of R_2_ and R_3_ with the R_1_ group = *S*-4-hydroxybenzyl as potentially being active agonists. These structures are shown and are denoted C1-S through C12-S. The R-stereoisomers at position R_1_ denoting C1-R through C12-R are not shown. (**B**) Selected dose–response curves in the indicated cells of two compounds, illustrating the screening process leading to further studies. C1-S responses were fit to a sigmoid curve (R^2^ > 0.95) with a Hill slope of 1.06 and this compound was taken forward to the biasing studies. In contrast, C7-S responses did not reach a plateau and this compound was not taken forward. Note also that C1-7 responses were not fully blocked by the β_2_AR antagonist propranolol (Figure 3C). (**C**) cAMP responses to C1-S through C12-S and C1-R through C12-R in the absence and presence of the receptor antagonist propranolol. C7-S and C8-S failed due to the lack of complete blockade of signaling by propranolol, while C2-S failed because its stereoisomer C2-R in the absence of propranolol was partially active (see text for details). *, *p* < 0.01 and †, *p* < 0.05, response to a compound is greater than baseline.

**Figure 4 jpm-12-00331-f004:**
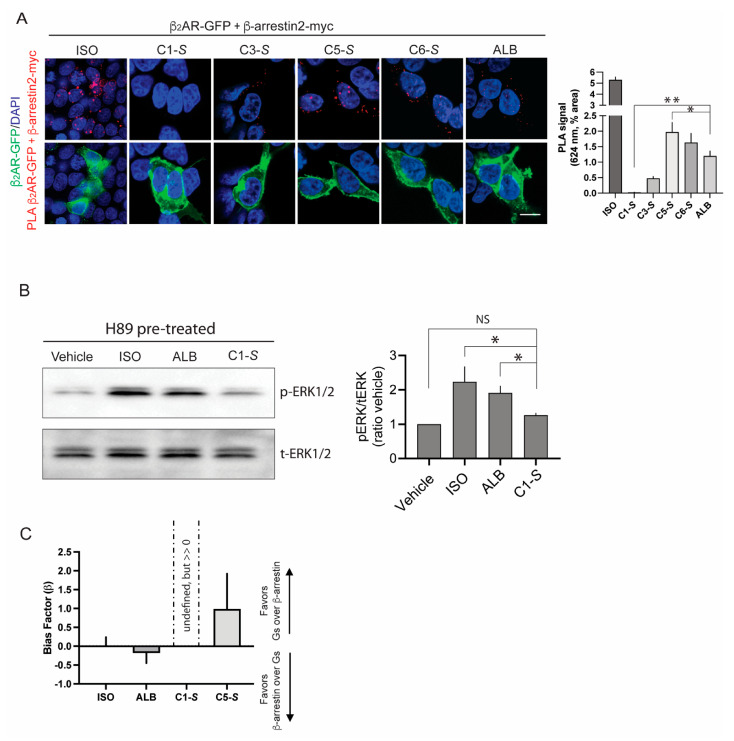
Biasing of C1-S away from β-arrestin. (**A**) Proximity ligation assay (PLA) in cells co-tranfected with tagged β_2_AR and β-arrestin2. In the representative confocal images, the red puncta indicate the agonist-promoted binding of β-arrestin2 to β_2_AR. The green signals are β_2_AR expressed on the cell surface, while the blue images are the nuclei stained with DAPI. The bar graph shows mean ± SE results from 5 independent experiments. C1-S at the highest concentration studied (300 µM) failed to evoke a PLA signal; * *p* < 0.05, ** *p* < 0.001, the PLA signal is different from albuterol. These results were confirmed in enzyme complementation assays and β-arrestin recruitment assays [11]. (**B**) C1-S failed to activate (phosphorylate) ERK1/2 in human ASM cells. A representative gel and results from 5 independent experiments are shown. Studies were performed in the presence of the protein kinase A inhibitor H89 in order to isolate β-arrestin function. * *p* < 0.05 versus vehicle control; NS, not significant. (**C**) Bias factors for the indicated agonists. The results from Gs-coupling and β-arrestin binding experiments were used to calculate the bias factor [35]. The results are shown as mean ±95% confidence intervals from 4 experiments. ISO, ALB, and C5-S had values of ~0 (or the confidence intervals crossed 0) and were considered balanced, in agreement with studies by others (see text). Since no β-arrestin interactions with β_2_AR could be evoked by C1-S, the bias factor is undefined (because of divisions by 0 in the formula), and thus by definition is >>0 and in favor of Gs over β-arrestin.

**Figure 5 jpm-12-00331-f005:**
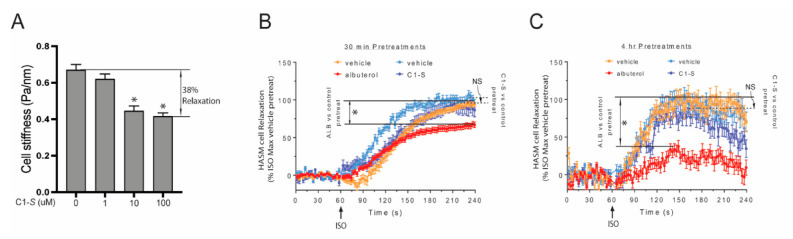
Physiologic confirmation of C1-S biasing away from β-arrestin and desensitization. (**A**) C1-S relaxes human ASM cells. Cell stiffness was measured my magnetic twisting cytometry, where a decrease in stiffness equates with relaxation. A ~38% relaxation response was observed with C1-S. *, *p* < 0.01. (**B**,**C**) C1-S, but not albuterol, fails to promote desensitization of the relaxation response to the full agonist isoproterenol (ISO). Human ASM cells were pretreated for 30 min or 4 h with vehicle (control), C1-S (100 μM), or albuterol (1.0 μM), washed, and then the relaxation response to isoproterenol (10 μM) was measured in real time. Albuterol pretreatment resulted in a ~35% (**B**) and ~70% (**C**) loss of ISO-evoked relaxation with the two pretreatment time points, respectively (*, *p* < 0.01). In contrast, C1-S incubation for either duration resulted in no statistically significant loss of β_2_AR function as assessed by ISO-evoked relaxation. NS = not significant.

**Figure 6 jpm-12-00331-f006:**
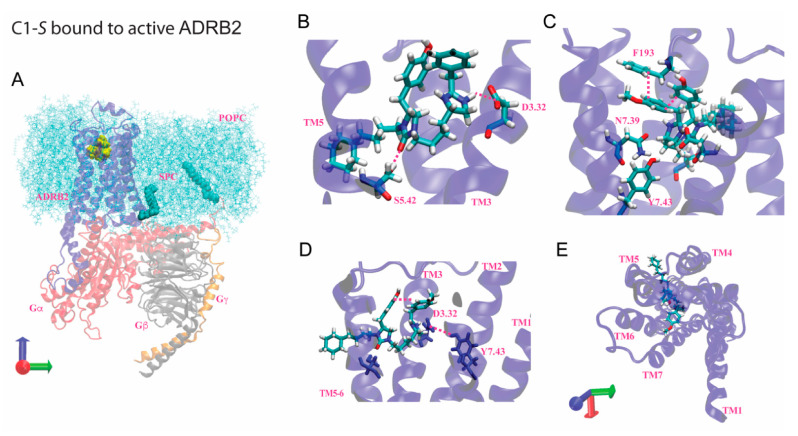
Predicted site for C1-S binding to β2AR coupled to Gs. (**A**) Orientation of C1-S (yellow) bound to the activated receptor (blue) in explicit membrane (light blue) and water. The G protein subunits are indicated in red (Gαs), grey (Gβ) and gold (Gγ). (**B**,**C**) Selected views of the interactions of C1-S with activated β_2_AR, including the imidazole interaction with Asp113^3.32^ (salt bridge) and Asn312^7.39^ (hydrogen bond), the urea interaction with Ser203^5.42^ (hydrogen bond), and π–π stacking of both aromatic rings of the agonist to each other and to Phe193^ECL2^. (**D**) The intrareceptor bond between Asp113^3.32^ and Tyr316^7.43^. (**E**) The TMD pocket as viewed from the extracellular side. POPC, phosphatidylcholine; SPC, palmitoylated region of the receptor and Gαs, which attaches to the membrane. See Figure 7 for the pharmacophores.

**Figure 7 jpm-12-00331-f007:**
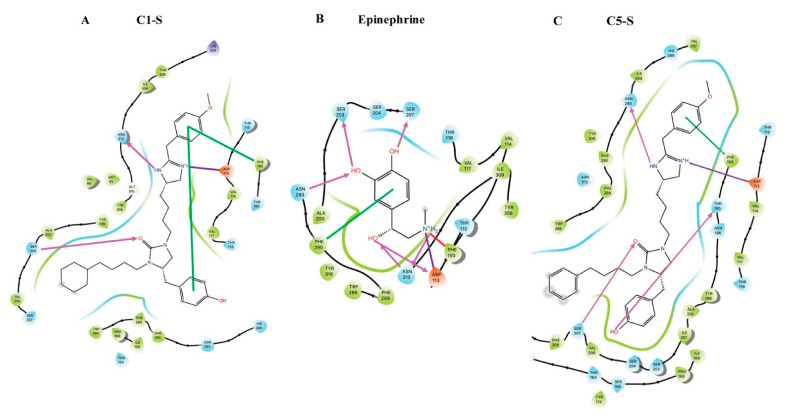
Pharmacophores showing the predicted binding sites for the agonists C1-S, epinephrine, and C5-S at the activated β_2_AR. See the text for comparisons. (**A**) C1-S binding sites include hydrogen bond (pink arrows) to Ser203^5.42^ and Asn312^7.39^; salt bridge (purple line) to Asp113^3.32^; and Pi–Pi stacking at Phe193^ECL2^ with an internal aromatic bond (green line). (**B**) Epinephrine binding sites include hydrogen bond to Ser203^5.42^, Ser207^5.46^, Asn293^6.55^, Asn312^7.39^ and Asp113^3.32^; salt bridge to Asp113^3.32^; cation–π interaction with Phe193^ECL2^; and a π-stacking interaction with Phe290^6.52^. (**C**) The C5-S binding site includes hydrogen bond to Ser203^5.42^, Asn293^6.55^, and Thr195^ECL2^, salt bridge to Asp113^3.32^; Pi-stacking at Phe193^ECL2^ with an internal aromatic bond. Ligand atoms that are exposed to solvent are marked with gray spheres.

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
