# Peer review of "Biased β-Agonists Favoring Gs over β-Arrestin for Individualized Treatment of Obstructive Lung Disease"

_jpm, 2022, doi:10.3390/jpm12030331_

Round 1
Reviewer 1 Report
Many review articles have been written on biased agonists of G protein-coupled receptors, which selectively activate β-arrestins. A review focusing on G-protein bias-type agonists, which has not been written before, is timely.
In the opiate receptor, it has been reported that the properties of G-protein biased agonists can be explained by low intrinsic activity (Kliewer et al., Br J Pharmacol 177, 2923-2931, 2020; Gillis et al., Sci Signal 13, eaaz3140, 2020; Gillis et al., Trends Pharmacol Sci 41, 947-959, 2020). This review describes a Gs protein biased agonist. In addition, this Gs biased agonist was a partial agonist. Therefore, it is possible that the new compound is not a Gs-biased agonist, but merely a partial agonist with very low activity. Considering these points, a new section should be devoted to the relationship between G-protein biased agonists and partial agonists.
Author Response
We thank the reviewer for the positive comments.
The one point that was made is related to partial agonists. We agree that this issue should be included in this Review Article and have inserted it starting at line 115 and going to line 126. And a new reference (one that was included in the reviewer’s comment) was cited, which is now reference #5.
Reviewer 2 Report
The authors of this manuscript describe, at least to my knowledge, this first Gs biased ligand for the β2 adrenergic receptor (recapitulated from Ref 11). This is a really nice achievement and will certainly be a valuable tool for the research community, if not a great led for novel medications in the treatment of asthma.
I was initially a bit confused about the statement of category "Review" and to rather find an article full of primary data with no adequate description of the methods. Some of the citations refer to the methods e.g. Ref 32 line 222 but are from a different receptor (formyl peptide receptor) and hence can´t contain the original data of the screen performed. Since the original data seem to be published in ref 11, I would suggest to cite this reference more often at the sections were the methods are mentioned. This way it will be easier for the readers to find the original data source...
Elsewise this article is timely and interesting.
Author Response
We thank the reviewer for the positive comments.
It was suggested that the primary reference for the example that we gave of a Gs-biased agonist, be provided when essential methods were in the text. This is reference #11, which has now been added in multiple locations in the manuscript as shown.
Round 2
Reviewer 1 Report
The author responds to the previous comment. No further comments.